# Near-Complete Remission of Glioblastoma in a Patient Treated with an Allogenic Dendritic Cell-Based Vaccine: The Role of Tumor-Specific CD4+T-Cell Cytokine Secretion Pattern in Predicting Response and Recurrence

**DOI:** 10.3390/ijms23105396

**Published:** 2022-05-12

**Authors:** Mariana P. Pinho, Guilherme A. Lepski, Roberta Rehder, Nadia E. Chauca-Torres, Gabriela C. M. Evangelista, Sarah F. Teixeira, Elizabeth A. Flatow, Jaqueline V. de Oliveira, Carla S. Fogolin, Nataly Peres, Analía Arévalo, Venâncio Alves, José A. M. Barbuto, Patricia C. Bergami-Santos

**Affiliations:** 1Department of Immunology, Instituto de Ciencias Biomedicas, Universidade de Sao Paulo, Sao Paulo 05508-000, Brazil; marianappinho@gmail.com (M.P.P.); emee.ruu@gmail.com (N.E.C.-T.); gabrielacoeli@usp.br (G.C.M.E.); sft.sarah@gmail.com (S.F.T.); eaflatow@gmail.com (E.A.F.); jaquelinevazdeoliveira@gmail.com (J.V.d.O.); carlafogolin@gmail.com (C.S.F.); nataly.peres@gmail.com (N.P.); 2Hospital das Clínicas HCFMUSP, LIM26, Faculdade de Medicina, Universidade de Sao Paulo, Sao Paulo 05403-000, Brazil; lepski@gmail.com (G.A.L.); analia.l.arevalo@gmail.com (A.A.); 3Department of Neurosurgery, Eberhard-Karls University, 72074 Tuebingen, Germany; 4Hospital do Coracao, Hcor, Sao Paulo 04003-905, Brazil; robertarehder1@gmail.com; 5Laboratory of Experimental Surgery (LIM-26), Hospital das Clínicas HCFMUSP, Faculdade de Medicina, Universidade de Sao Paulo, Sao Paulo 05403-000, Brazil; 6Department of Pathology, Faculdade de Medicina, Universidade de Sao Paulo, Sao Paulo 01246-903, Brazil; venancio@uol.com.br; 7Laboratory of Medical Investigation in Pathogenesis and Targeted Therapy in Onco-Immuno-Hematology (LIM-31), Department of Hematology, Hospital das Clínicas HCFMUSP, Faculdade de Medicina, Universidade de Sao Paulo, Sao Paulo 05403-000, Brazil

**Keywords:** glioblastoma, immunotherapy, dendritic cells, CD4+ T cells, vaccine, cancer treatment

## Abstract

Immunotherapy has brought hope to the fight against glioblastoma, but its efficacy remains unclear. We present the case of CST, a 25-year-old female patient with a large right-hemisphere glioblastoma treated with a dendritic–tumor cell fusion vaccine. CST showed a near-complete tumor response, with a marked improvement in her functional status and simultaneous increases in tumor-specific CD8+ and CD4+ T cells. Two months before recurrence, the frequency of tumor-specific T cells decreased, while that of IL-17 and CD4+ T cells increased. CST passed away 15 months after enrollment. In this illustrative case, the tumor-specific CD4+ T-cell numbers and phenotype behaved as treatment efficacy biomarkers, highlighting the key role of the latter in glioblastoma immunotherapy.

## 1. Introduction

Glioblastoma accounts for 47% of malignant intracranial tumors [1]. The gold standard treatment for glioblastoma consists of maximal surgical resection followed by concomitant and adjuvant radio- and chemotherapy with temozolomide [2]. However, the survival rate remains dismal, with an overall rate of 14.6 months [2].

One recent approach is immunotherapy [3,4,5,6] involving dendritic cells (DCs). DCs can drive both tolerance and immunity [7,8], since they are the gateway through which antigens are recognized by T cells in the immune system. Not surprisingly, therefore, DCs have been increasingly recognized as defective in cancer patients, and potentially effective tools for immunotherapy [9]. Most recently, their use has yielded promising results among patients with newly diagnosed glioblastoma [10], and there are many clinical trials underway attempting to harness their potential for activating the immune system to fight glioblastoma [11] For clear immunological reasons, most of these studies use autologous DCs, since antigens are recognized in the context of HLA molecules, which are extremely heterogeneous among individuals. However, cancer patients’ DCs are frequently biased towards tolerance, and may, thus, fail to induce fully effective immune responses [12,13]. This obstacle may be circumvented by using DCs from healthy donors fused with autologous tumor cells [14,15], as described here, in a strategy that bypasses the potential tolerogenic bias of cancer patients’ DCs and can also add an allogeneic effect to the DC-induced immune response [16].

Here, we present the case of CST, a young woman admitted to palliative care who showed near-total remission of glioblastoma over a 12-month period of vaccination, with allogenic DCs fused with autologous tumor cells.

## 2. Case Report

In November 2014, CST, a 25-year-old previously healthy woman, underwent surgery for a right frontal anaplastic astrocytoma, World Health Organization (WHO) grade III, IDH-1 mutated (R132H), non-methylated MGMT, and 1p19q non-codeleted. Following surgery, CST was submitted to 30 sessions of photon intensity-modulated radiotherapy (total 60 Gy) with concurrent (75 mg/m^2^) and adjuvant (4-week regimen; 150 mg/m^2^ given on days 1–5) temozolomide for 6 months [17].

The tumor recurred as a glioblastoma in December 2017, and a second surgical resection was performed in March 2018, followed by re-irradiation (60Gy, completed in May 2018). In August 2018, the tumor relapsed, with a significant mass effect of a contrast-enhancing component with extensive necrosis in the right frontal lobe (Figure 1). At the neurological exam, CST was alert and conscious (GCS 15), with left spastic hemiparesis (grade III) and no cognitive or language deficits. She was able to take care of herself most of the time (KPS 60). In September 2018, CST enrolled in a phase II clinical trial for an autologous tumor-allogeneic DC fusion vaccine (National Research Council approval Nr. 58882116.7.3001.0065).

The vaccine was administered monthly for 12 months. MRI scans were acquired every two months, and the response to treatment was evaluated according to RANO criteria [18]. During the vaccination period, CST continued to receive bevacizumab every 2 weeks (10 mg/Kg), which she already received prior to inclusion in the study. The leucocyte counts remained between 4190 and 5080 cells/µL, and the T-lymphocyte counts between 1292 and 1508 cells/µL. Figure 2 shows the MRI scans and functional scale scores throughout the period. Before study admission, and 30 days after the last surgical intervention, CST presented with MMSE 23/30, KPS 50/100, ECOG 3, and progressive disease, according to RANO criteria. The FLAIR sequence showed an extensive tumor affecting most of the right hemisphere, with a collapsed right frontal horn and midline shift to the left.

On vaccination days, the circulating T cells were analyzed (Figure 3A). Before vaccination, the frequency of circulating tumor-specific CD4+ and CD8+ T cells in CST’s blood was below assay detection levels (Figure 4A). After in vitro expansion, tumor-specific T-cell subpopulations were identified by a Uniform Manifold Approximation and Projection (UMAP) analysis (Figure 3B–D), where we noticed an increased frequency of tumor-reactive Tregs and a lower frequency of Th1 cells, relative to polyclonally activated cells (Figure 4E).

At the time of the second dose, CST’s MMSE and KPS improved to 25/30 and 60/100, respectively, but her ECOG remained at 3. She reported slight improvements in the quality of life and global health scales (Figure 1). The MRI showed tumor shrinkage with almost no mass effect (Figure 1). The immunological analysis four weeks later showed a striking increase in the frequency of circulating tumor-specific CD4+ T cells, a smaller, but detectable, increase in tumor-specific CD8+ T cells (Figure 2), an increase in Th1 cells, and a reduction in all other subpopulations (Figure 4B,C).

CST continued to improve (Figure 2), with clinical deterioration occurring only after the 10th dose. Initially, the FALIR hyperintensity and contrast-enhancing spots increased slightly. Three months prior, the total number of tumor-specific CD4+ T cells and Th1 cells decreased, while Th2 increased. Two months thereafter, IL-17-producing CD4+ T cells became the predominant tumor-specific subpopulation (Figure 2). Though tumor-reactive Tregs increased, their frequency did not reach pre-therapy levels (Figure 4E).

The last MRI (acquired at the 12th dose) revealed a large tumor mass occupying both frontal lobes and exerting significant mass effects. CST presented with global aphasia, right hemiparesis (grade II), a KPS score of 50, and an ECOG score of 3. Her mother requested that we discontinue vaccination. CST passed away in January 2020, due to tumor progression, 15 months after study enrollment. Her overall survival was 63 months from diagnosis.

## 3. Discussion

CST’s immune response profile during the vaccine treatment suggests a critical role of CD4+ T cells in immunotherapy response and failure. At first, this finding may seem to be at odds with the most prevalent current view, which emphasizes the critical role of CD8+ T cells. CD8+ T cells play an unequivocal role in recognizing and responding to changes in the molecular configuration of any nucleated cell in the body. However, a growing amount of data obtained in studies of both CD4+ and CD8+ T cells indicate that the role of CD4+ may be at least as important as that of CD8+ T cells [21]. Despite these findings, the role of CD4+ is frequently ignored and understudied. In the patient presented here, besides preceding the appearance of tumor-specific CD8+ T cells, the increase in CD4+ tumor-specific T cells and their cytokine secretion pattern predicted CST’s clinical outcome. The association between the Th17 response and clinical degradation highlights the need to further investigate these cells’ roles in glioblastoma immune control and escape.

Differently from other immunotherapy approaches, vaccines can induce new immune responses. To optimize efficacy, vaccines should focus on exploiting the unique ability of DCs to change tolerance into immune-responsive states [22,23]. In the study conducted by Liau et al., patients with glioblastoma, vaccinated with autologous DCs pulsed with tumor lysates (DCVax), had 2- and 3-year survival rates of 46.2% and 25.4%, respectively [10].

Differently from previous studies, we used allogeneic monocyte-derived DCs (mo-DCs) to bypass cancer patients’ mo-DCs bias toward Treg induction [12,24]. To circumvent the need for HLA-compatible monocyte donors, we electrofused the tumor and DCs, generating effective antigen-presenting cells [25] with the immunity-enhancing allogeneic effect [16].

We observed the effectiveness of this strategy in CST’s remarkable response. Al-though she showed very low numbers of tumor-specific T cells and a high Treg frequency before immunotherapy, there was a clear shift in her response pattern following the first dose; the total frequency of circulating CD4+ T and Th1 cells increased, while that of Tregs decreased, as did the frequency of Th1/2, Th2, Th1/17, and Th17 subpopulations. This immune scenario proved to be a predictor of CST’s positive clinical evolution. Although we cannot exclude a contributing role for bevacizumab, it is difficult to attribute this drug to the immunological cascade of events and clinical–radiological changes that occurred later, when the vaccination was started. It remains to be determined, however, whether bevacizumab had an additional, synergistic effect together with the vaccination. Since bevacizumab is incorporated in guidelines for glioblastoma recurrence, and its efficacy alone has been documented in many trials, we would have had no ethical endorsement to discontinue its usage.

Although we were not surprised to detect high frequencies of tumor-specific Tregs and Th2 cells prior to vaccination, the relatively high frequency of Th1/17 and Th17 was unexpected. Th17 cells have been linked to both anti- and pro-tumoral activity, and their role is still poorly understood [26,27]. Interestingly, CD8+ T cells appeared later, and did not correlate as well with CST’s clinical profile. This could be related to their concentration within the tumor itself, with the circulating cells merely reflecting an overflow of tumor-infiltrating cells. The frequency and cytokine secretion pattern of circulating CD4+ T cells, however, were better predictors of CST’s clinical evolution.

CST maintained a favorable clinical and immune response to vaccination for six months. She then showed a brief surge in Th2 activity, followed by an increase in IL-17-producing cells, and clinical deterioration three months later. The observation that Th1/17 increased more than Th17 cells suggests that Th1 cells may have been converted into Th1/17 cells, losing their anti-tumor effectiveness, and allowing tumor escape. We do not have enough data to explain these subpopulation fluctuations. Speculatively, though, we could suggest that they may be coherent with the immune system’s physiology, where continuous stimuli lead to immune response pattern shifts towards less destructive patterns. In CST’s evolution, this would have been represented by the surge in the tumor-specific Th2 response. Contrary to infectious diseases—where such pattern modifications are frequently accompanied by a parasite adaptation and consequent decrease in tissue destruction— the tumor, which is not subjected to an evolutionary pressure to allow the host’s survival, does not “stop”. In this context, we could speculate further that by facing continuous aggression, a subsequent “destructive” immune response was recruited—the Th17 response. Undoubtedly, future work should elucidate which fluctuations contribute to tumor escape and how to modify treatments to avoid this.

CST survived for 63 months with a disease that has a mean overall survival of 19.8 ± 2.5 months [28], thus presenting a remarkable clinical response to an allogeneic DC-autologous tumor fusion vaccine. Her immunological evolution highlights the need to also focus on CD4+ T-cell response patterns in patients submitted to immunotherapy. Ultimately, defining an optimal CD4+ T-cell and CD8+ T-cell response pattern, and developing ways of keeping the CD4+ T-cell response pattern within boundaries for longer periods, should help optimize immunotherapeutic approaches against glioblastoma and other neoplastic diseases.

## 4. Methods

### 4.1. Patient Recruitment and Ethics

CST was the first participant enrolled, following written informed consent, in the phase II prospective trial on DC vaccination for glioblastoma. All procedures were approved by the Institutional Ethics Committee and National Research Council (approval 58882116.7.3001.0065). Patient data were collected prospectively, anonymously, and recorded using RedCap (https://redcap.hc.fm.usp.br, accessed on 20 February 2021).

### 4.2. Follow-Up

Neurologic status was assessed monthly by a general neurologic exam and the Mini-Mental Status Exam. Overall performance was assessed by the Karnofsky performance status (KPS) and WHO-ECOG, whereas global health and quality of life were assessed by the EORTC-QLQ-C30, EORTC-QLQ-BN20, FACT-Br, and MDASI-BT evaluation scales. MRI scans were obtained every 2 months, and tumor progression was defined according to RANO criteria. Adverse events were categorized according to the US NCI’s Common Terminology Criteria for adverse events version 4.0.245c.

### 4.3. Tumor Sample Processing and Vaccine Production

Dendritic–tumor cell hybrids were obtained as described previously [6]. Briefly, the fresh tumor sample, obtained from surgical resection of the secondary GBM lesion, was digested and the single-cell suspension was kept in liquid nitrogen. Activated mo-DCs were generated from different donors for each dose and electrofused with freshly thawed tumor cells. Hybrid cells were irradiated (200 Gy) before intradermal injection.

### 4.4. Tumor-Reactive T-Cell Frequency and Phenotype

The frequency of tumor-reactive T cells was calculated as described [19]. Briefly, T cells and autologous mo-DC were co-cultured at a 10:1 ratio and tumor lysates were added to 10 replicate wells (10 μg of protein/mL). After five days, the number of wells without proliferating T cells was used to calculate the tumor-reactive T-cell frequency.

Proliferating T cells (CFSElowCD25+CD4+T cells) were sorted and expanded in the presence of allogeneic PBMC, phytohemagglutinin, IL-2, IL-7, and IL-15. After challenge with PMA, ionomycin and brefeldin were stained (live/dead marker, CD4, CD8, IFN-γ, IL-4, IL-10, and IL-17) and analyzed.

## Figures and Tables

**Figure 1 ijms-23-05396-f001:**
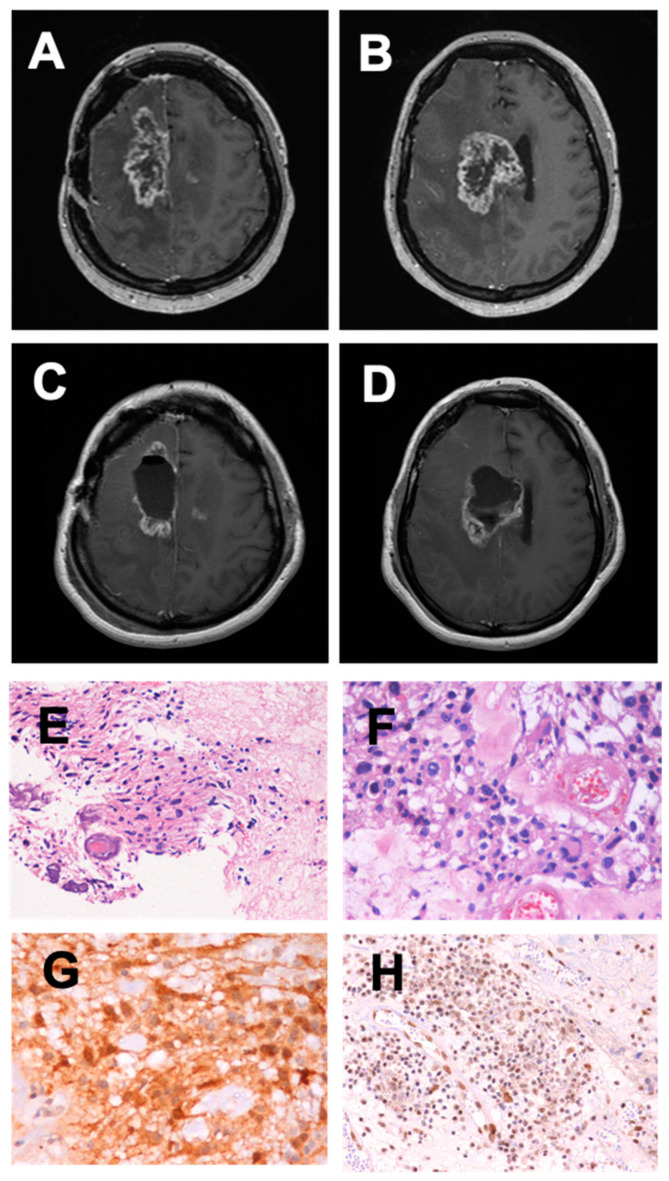
MRI scans and histopathology of tumor. MRI scans were obtained immediately before and after the last surgical intervention and prior to study admission. In (**A**,**B**), T1-weighted sequences with gadolinium show the contrast-enhancing tumor components in the right frontomedial cortex and deep right frontal lobe, causing a significant mass effect. At this time (September 2018), the patient was in KPS60, and a new surgical evacuation of the tumor was indicated. In (**C**,**D**), T1-weighted sequences with gadolinium show post-operative results. CST was then referred to the vaccination study. (**E**) Hematoxylin–eosin staining showing highly cellular and pleomorphic infiltrative neoplasm and depicting spindled, ovoid, or stellate glial cells with marked anaplasia (200× magnification). (**F**) Hematoxylin–eosin staining at higher magnification (400×), where proliferated microvessels are depicted amidst marked pleomorphic glial cells. (**G**) Immunohistochemical staining for IDH-1 mutation, where neoplastic cells present strong and diffuse nuclear and cytoplasmic expression of IDH-1 (400×). (**H**) Immunohistochemical staining for ATR-X showing preserved nuclear expression (200×).

**Figure 2 ijms-23-05396-f002:**
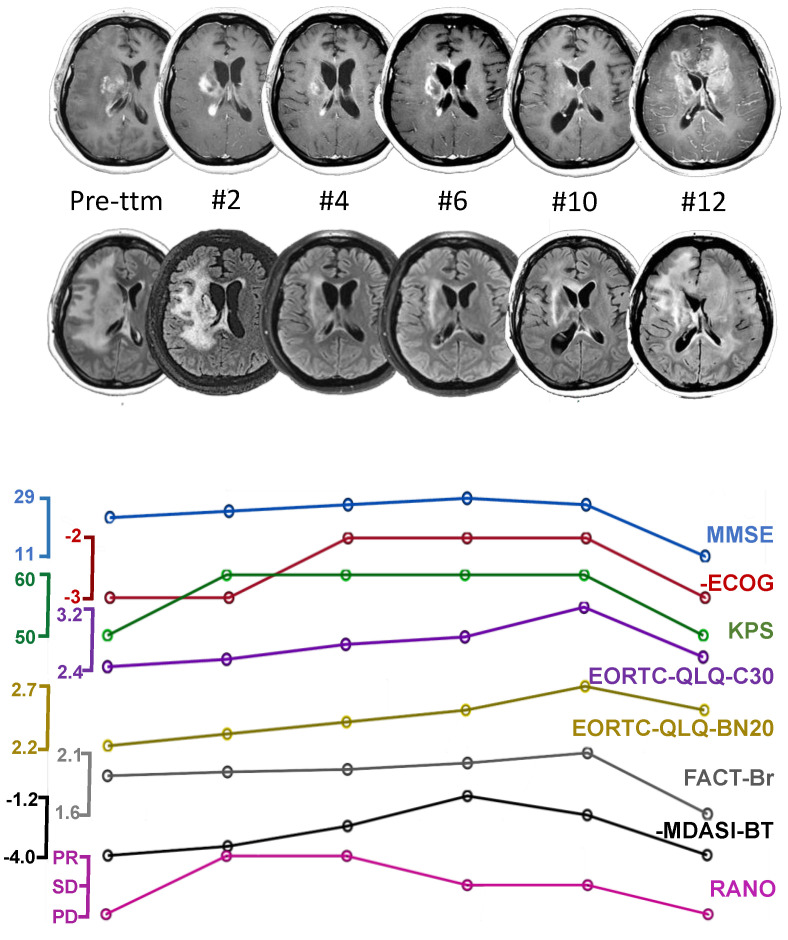
Clinical radiological outcome during vaccination. First line: T1-weighted MRI images with gadolinium. From left to right: images obtained over progressive timepoints, starting with pre-treatment. Numbers below each image correspond to the vaccination dose number, as well as the month post-treatment when each dose was taken. Second line: corresponding FLAIR images obtained at each vaccination dose. Note the significant tumor reduction beginning at dose #4 observed on both sets of images, with significantly reduced midline shift and mass effect (most visible on FLAIR), followed by a stable period from dose #6 to #10, and then rapid deterioration from dose #10 to #12. The line graphs represent the evolution of clinical and quality of life scores over the 12 months of vaccination. Each point is temporally matched to the images above. Score signs were adjusted so that upward evolution indicates clinical improvement. MMSE improved consistently from 23 at the beginning to 25 by dose #2, 27 by dose #4, and 29 by dose #6. At dose #2, CST’s KPS had improved by 10 points and remained at that level until dose #10; by dose #4, ECOG improved by 1, and remained stable up to dose #10. QoL according to EORTC scores, FACT-Br, and MDASI improved progressively, while RANO criteria indicated a partial response corresponding to tumor shrinkage. After dose #10, progressive clinical and radiological deterioration was observed for all measures. Legends: PR: partial response, SD: stable disease, PD: progressive disease.

**Figure 3 ijms-23-05396-f003:**
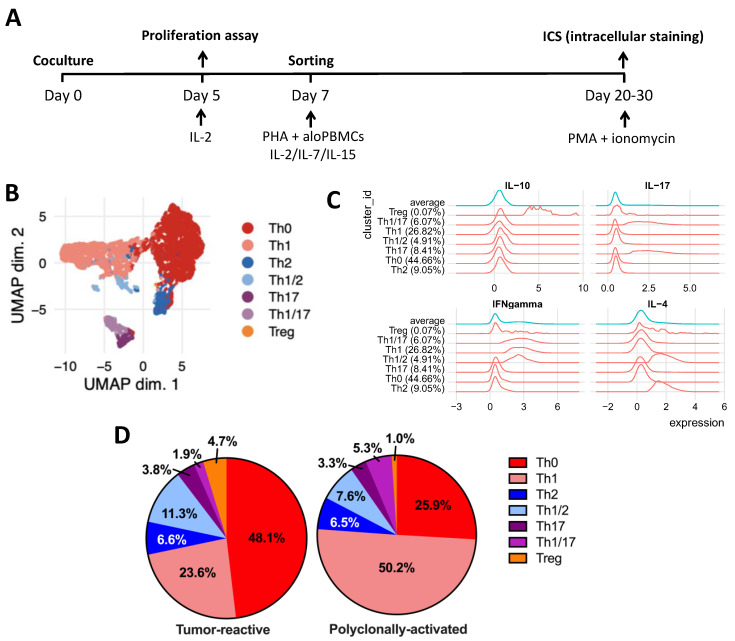
Characterization of CD4+ Th cell subpopulation using bioinformatic tools. (**A**) Experimental design to evaluate frequency and phenotype of tumor-reactive T cells in CST’s blood. Blood was collected before each vaccination dose, and PBMCs were obtained by gradient centrifugation. CFSE-labeled T cells were cocultured, in replicates, with monocyte-derived dendritic cells in the presence of tumor lysate for 5 days, when half of each well was independently assessed for proliferation by flow cytometry to calculate the frequency of tumor-reactive T cells. The other half of each well and the replicates not used for the proliferation assay were cultured in the presence of IL-2 for 2 more days and the CFSE^low^CD25^+^ proliferating T cells were sorted. Sorted cells were expanded in the presence of PHA, irradiated allogeneic PBMCs (allo-PBMC), IL-2, IL-7, and IL-15. After expansion, the cells were stimulated overnight with PMA + ionomycin in the presence of brefeldin, and their cytokine production profile was assessed via intracellular staining (ICS) [19]. (**B**) The identification of the different T-helper subtypes was conducted by means of unsupervised clustering using FlowSOM and ConsensusClusterPlus R packages, with the ICS flow cytometry data of expanded tumor-reactive T cells [20]. The cell subtypes were visualized using the dimension reduction technique, UMAP. (**C**) Histograms of cytokine expression per cluster obtained by the FlowSOM clustering. (**D**) CD4+ T cells proliferating to a polyclonal CD3/CD28 bead polyclonal stimulus or to the presence of antigen-presenting cells loaded with the patient’s tumor lysate (tumor-reactive) were sorted, expanded, and had their helper phenotype determined (as described above). The frequency of each population in the patient’s blood at the onset of immunotherapy was plotted as a percentage in the pie chart.

**Figure 4 ijms-23-05396-f004:**
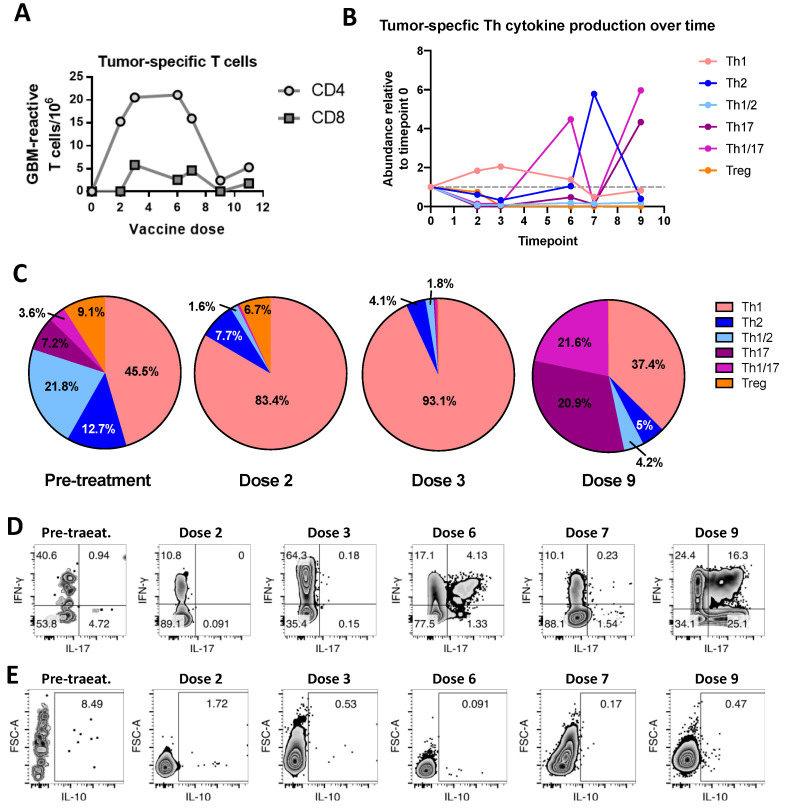
Evolution of tumor-specific CD4+ T-cell phenotype during treatment. (**A**) The frequency of CD4+ and CD8+ tumor-specific T cells in the patient’s blood was determined by evaluating the presence of proliferating T cells, when challenged with the tumor lysate, in independent wells. The calculation was performed using the number of negative wells according to the Poisson distribution [19]. (**B**) We obtained blood from the patient at different timepoints (corresponding to the number of vaccine doses received), and tumor-reactive CD4+ T cells were isolated and expanded. The frequency (among cells producing at least one cytokine, thus excluding the Th0 population) was plotted relative to timepoint 0. (**C**) The proportion of cells in each Th population was plotted in pie charts for timepoints 0 (pre-treatment), 2, 3, and 9. (**D**) Dot plots of IFN-γ versus IL-17 expression by GBM-reactive CD4+ T cells in the blood of the patient at different treatment timepoints. (**E**) Dot plots of IL-10 expression by GBM-reactive CD4+ T cells in CST’s blood at different treatment timepoints.

## Data Availability

All clinical and laboratorial data were collected according to the national ethical rules and legislation, and are available for analysis upon request.

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
