# Peer review of "Near-Complete Remission of Glioblastoma in a Patient Treated with an Allogenic Dendritic Cell-Based Vaccine: The Role of Tumor-Specific CD4+T-Cell Cytokine Secretion Pattern in Predicting Response and Recurrence"

_ijms, 2022, doi:10.3390/ijms23105396_

Round 1

Reviewer 1 Report

This manuscript is strictly related to clinical medicine and its molecular aspect is very limited as it essentially deals with two parameters related to the T cells. In general, this is a well written manuscript clearly presenting all its aspects, but as a case report, its novelty cannot be generalized. Moreover, the conclusion on the key role of CD4+Tcell phenotype in glioblastoma immunotherapy is unjustified. This manuscript should be submitted to a more clinically-oriented journal, e.g., MDPI Journal of Clinical Medicine or Cancers and not to IJMS.

Author Response

Please, see attachment.

Reviewer 2 Report

In this manuscript, the authors report on the use of tumor-activated allogenic dendritic cells against glioblastoma. They provide a thorough assessment of the T cell immunophenotype and suggest that the T cell immunophenotype could be used as a biomarker, providing useful insights to the treatment progression and prognosis.

The manuscript is well-written and the findings are clearly presented, in a logical continuum. My only suggestion would be to elaborate more on the mechanisms underlying the switch in the T cells immunophenotype, which appears to eventually lead to tumor evasion.

Author Response

Please, see attachment.

Reviewer 3 Report

In this manuscript, the authors present a clinical case in which they have achieved a significant reduction of a glioblastoma by immunotherapy with the use of a dendritic-tumor cell fusion vaccine over a 12-month period of vaccination.

The manuscript is well written and is easy to read. It also presents a very interesting case. I would like to recommend the publication of this manuscript and I have only a number of questions that I think would be interesting and that have not become clear to me after reading the manuscript:

  • What effect could the fact that the patient continued to receive simultaneous treatment with Bevacizumab have had on the results of this clinical trial?

  • What could account for the change, 6 months after the start of vaccination, in the tendency to decrease certain cell populations related to CTS cancer and the increase in others of these populations?

  • If treatment had not been interrupted at dose 12, do the authors consider that the trend of the populations could have changed back to a more favourable one? Do the authors consider that the worsening would have been increased or reduced by continuing treatment?

Author Response

Please, see attachment.

Reviewer 4 Report

Title of the manuscript: Near-complete remission of glioblastoma in a patient treated with an allogenic dendritic cell-based vaccine: the role of tumor-specific CD4+T-cell cytokine secretion pattern in predicting response and recurrence

By Patricia C. Bergami-Santos, Guilherme A. Lepski, Mariana P. Pinho, Roberta Rehder, Nadia E. Chauca Torres, Gabriela C. M. Evangelista, Sarah F. Teixeira, Elizabeth A. Flatow, Jaqueline V. de Oliveira, Carla S. Fogolin, Nataly Peres, Analía Arévalo, Venâncio Alves, and José A. M. Barbuto

The manuscript is a Case Report article on the results of the treatment of a 25 y.o. woman initially diagnosed with grade III right frontal anaplastic astrocytoma, who was operated and treated with radiotherapy and temozolomide. Three years later, the patient developed a secondary glioblastoma and was treated with second surgery and radiotherapy. After the relapse of the GBM, the patient was vaccinated  with allogenic dendritic cells fused with autologous tumor cells and showed near-total remission of the tumour & great functional improvement paralleled with increase of the numbers of the tumor-specific CD8+ and CD4+T-cells during 12 months of vaccination. However, the patient developed a recurrence later and died 15 months after the start of the involvement in the vaccination study (63 months after the first brain tumour diagnosis). The patient was CST enrolled in a phase II clinical trial for an autologous tumor-allogeneic DC fusion vaccine (National Research Council approval Nr. 7758882116.7.3001.0065). There was an improvement in MRI and life quality results from 2nd to 10th month of the monthly vaccine dosing and rapid deterioration after the doses #10 to #12. During the vaccination period and before inclusion in the srudy, the patient was administered Bevacizumab every 2 weeks. The improvement was marked with the increase of CD4+ and CD8+ T cells, while deterioration was associated with the decrease of the number of  CD4+T-cells and Th1 cells, increase of  Th2 cells and predominance of IL-17-producing CD4+T-cells. The study indicated an important role of CD4+T-cells in GBM immunotherapy response and the link between Th17 response and tumour immune escape.

To the best of my knowledge, the manuscript presents the results of a well organised and ethically acceptable clinical trial. I think that the reported findings are very important for better understanding of the dendrite cell vaccination approaches in GBM and further improvement of the clinical trial design. The results and methodology parts are well written and comprehensive. I have only minor suggestions that may improve the readability of the manuscript:

  1. Please, provide a more detailed introduction on the topic (DC vaccination in glioma patients). What is the scientific background of this approach, what has been done and what are the current achievements and challenges? This would help to better understand the position of the current study in the state-of-the-art treatment of brain malignant tumours.
  2. Please, discuss the overall outcome of the treatment with DC vaccine vs the current GBM survival data. It is not completely clear whether a survival extension was achieved or not, or whether the functional status improvement was clinically significant.
  3. Please, enhance and deepen the discussion & mechanistic interpretation of the changes of the T cell phenotypes during the DC vaccine application and the recurrence of the disease.
  4. Please, correct the spelling of the types of the cells using superscripts and subscripts for the specific phenotypes.
  5. Please, provide a larger version of Figure 1 and the individual images used in this panel.
  6. In the Figure 1 annotation, please, check if you missed any labelling on the histological image or correct the grammar in the phrase “where proliferated microvessels are depicted amidst marked pleomorphic glial cells”.

Author Response

Please, see attachment.

Round 2

Reviewer 1 Report

I sustain my previous opinion.